# Comparing Simulations of Umbrella-Cloud Growth and Ash Transport with Observations from Pinatubo, Kelud, and Calbuco Volcanoes

**Larry G. Mastin *** 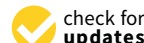 **and Alexa R. Van Eaton**

U.S. Geological Survey, David A. Johnston Cascades Volcano Observatory, 1300 SE Cardinal Court, Bldg. 10, Suite 100, Vancouver, WA 98683, USA; avaneaton@usgs.gov

*** Correspondence: lgmastin@usgs.gov; Tel.: +1-360-993-8925

**Abstract:** The largest explosive volcanic eruptions produce umbrella clouds that drive ash radially outward, enlarging the area that impacts aviation and ground-based communities. Models must consider the effects of umbrella spreading when forecasting hazards from these eruptions. In this paper we test a version of the advection–dispersion model Ash3d that considers umbrella spreading by comparing its simulations with observations from three well-documented umbrella-forming eruptions: (1) the 15 June 1991 eruption of Pinatubo (Philippines); (2) the 13 February 2014 eruption of Kelud (Indonesia); and (3) phase 2 of the 22–23 April 2015 eruption of Calbuco (Chile). In volume, these eruptions ranged from several cubic kilometers dense-rock equivalent (DRE) for Pinatubo to about one tenth for Calbuco. In mass eruption rate (MER), they ranged from $10^8$–$10^9$ kg s$^{-1}$ at Pinatubo to 9–16 × $10^6$ kg s$^{-1}$ at Calbuco. For each case we ran simulations that considered umbrella growth and ones that did not. All umbrella-cloud simulations produced a cloud whose area was within ~25% of the observed cloud by the end of the eruption. By the eruption end, the simulated areas of the Pinatubo, Kelud, and Calbuco clouds were 851, 53.2, and $100 \times 10^3$ km$^2$ respectively. These areas were 2.2, 2.2, and 1.5 times the areas calculated in simulations that ignored umbrella growth. For Pinatubo and Kelud, the umbrella simulations provided better agreement with the observed cloud area than the non-umbrella simulations. Each of these simulations extended 24 h from the eruption start. After the eruption ended, the difference in cloud area (umbrella minus non-umbrella) at Pinatubo persisted for many hours; at Kelud it diminished and became negative after 14 h and at Calbuco it became negative after ~23 h. The negative differences were inferred to result from the fact that non-umbrella simulations distributed ash over a wider vertical extent in the plume, and that wind shear spread the cloud out in multiple directions. Thus, for some smaller eruptions, wind shear can produce a larger cloud than might be produced by umbrella spreading alone.

**Keywords:** volcanic ash clouds; umbrella clouds; explosive volcanic eruptions; transport and dispersion models; aviation meteorology; Pinatubo; Kelud; Calbuco

## 1. Introduction

Numerical models are essential for forecasting tephra transport during eruptions. For small to medium-sized eruptions (smaller than about 4 on the Volcanic Explosivity Index (VEI) scale [1]), most models advect ash downwind in the ambient wind field and ignore any winds created by the eruption itself. Larger eruptions, however, produce umbrella clouds with radial winds that can advect ash in crosswind and upwind directions. As eruption rate increases, radial spreading becomes increasingly important. While small eruptions produce a fan-shaped cloud in map view, widening downwind from a point source, large eruptions such as Pinatubo in 1991 produce a circular cloud.

Umbrella spreading has been incorporated into at least three numerical models used for ash-cloud dispersal: the Numerical Atmospheric-dispersion Modeling Environment (NAME) model [2], Fall3d [3], and Ash3d [4]. Validation studies using NAME [2] have compared modeled clouds with observed ones during historical eruptions, and a study using Fall3d [5] has compared modeled versus measured deposit thickness for the ancient Toba supereruption. No validation studies have yet been reported for Ash3d.

Ideally, an umbrella-spreading algorithm should faithfully simulate eruptions both small and large, with and without umbrellas. A validation study of such a model should therefore test modeled cloud growth over a wide range of eruption sizes. Our objective is to test an umbrella cloud implementation in the U.S. Geological Survey's (USGS) Ash3d model using three eruptions whose mass eruption rate and erupted volume ranged over about two orders of magnitude: (1) the 15 June 1991 Pinatubo eruption in the Philippines, in which a 9-hour-long VEI 6 eruption with mass eruption rate (MER) of ~$10^8$–$10^9$ kg s$^{-1}$, produced a circular cloud about $1200 \times 1400$ km$^2$ in size [6,7]; (2) the 13 February 2014, Kelud eruption in Indonesia, whose MER of $0.3$–$1.3 \times 10^8$ kg s$^{-1}$ produced an oval umbrella $200 \times 300$ km$^2$ in size [8]; and (3) the 22–23 April 2015 eruption of Calbuco volcano in Chile, whose two phases erupted at MER = ~$0.6$–$1.6 \times 10^7$ kg s$^{-1}$ and produced umbrella clouds up to $100 \times 160$ km$^2$ in size [9,10].

## 2. Dynamics of Umbrella Growth

Umbrella clouds are gravity currents produced when a volcanic column rises to a top height $H_T$ beyond its level of neutral buoyancy ($H_{NBL}$), collapses to a lower elevation, then oscillates around $H_{NBL}$ as it moves radially outward and/or downwind. In small eruptions or high winds, the ash is advected downwind as it oscillates, producing a weak plume (Figure 1a). In large eruptions and/or weak winds, the collapsing plume expands radially, driving ash in both crosswind and upwind directions (Figure 1b). The radially expanding cloud oscillates in elevation, entrains air, and finally settles at an elevation $H_u$, at or slightly above $H_{NBL}$ (Figure 1b).

The importance of radial spreading depends on wind speed, MER, and eruption duration ($D$). The gravitational head that drives spreading is resisted by air drag. Assuming constant MER, a simple force balance [11] suggests that the cloud radius $R$ grows with time $t$ in proportion to $t^{2/3}$ [3,11,12]. Conceptual models of umbrella growth predict time-varying values of the time exponent [13], but several well-documented umbrella clouds fit a $R \propto t^{2/3}$ curve well [9,14–17]. Thus, the umbrella-cloud growth rate is commonly expressed as [3]

$$R = \left(\frac{3\lambda NQ}{2\pi}\right)^{1/3} t^{2/3}, \tag{1}$$

where $\lambda$ is an empirical shape constant estimated at 0.1 to 0.6 [12,17], $N$ is the Brunt-Väisälä frequency, and $Q$ is the volume influx rate of ash, gas and entrained air into the cloud. Several studies [2,4,9,18] have used $\lambda \approx 0.2$, following Costa, et al. [3] or Rooney and Devenish [12]. From three-dimensional (3D) plume modeling, Suzuki and Koyaguchi [17] found that values of 0.2 and 0.1 work best for tropical and mid-latitude or polar conditions, respectively, and we follow that recommendation. The $Q$ is calculated using a corrected version of Equation (2) from Costa, et al. [3] (see Appendix A for the uncorrected version):

$$Q = C \sqrt{k_e} \frac{\dot{M}^{3/4}}{N^{5/4}} \tag{2}$$

where $\dot{M}$ is the mass eruption rate, $k_e$ is the entrainment coefficient into the plume, and $C$ is an empirical constant, equal to approximately 430 m$^3$ kg$^{-3/4}$ s$^{-3/2}$ for tropical eruptions and 870 m$^3$ kg$^{-3/4}$ s$^{-3/2}$ for mid-latitude and polar eruptions.

The cloud's outer edge travels outward with time at a speed $dR/dt = u_R$. Within the cloud, a radial wind speed $u(r)$ (Figure 1b) decreases with $r$ so that $u(r) \to u_R$ as $r \to R$. Umbrella clouds tend

to flatten and thin as they expand. Assuming that $u_R = \lambda N h$, the rate of flattening can be derived as $(dh/dt) = -Q/(3\pi R^2)$. From this, Costa, et al. [3] derived the function

$$u(r) = u_R \left( \frac{3}{4} \frac{R}{r} + \frac{1}{4} \frac{r}{R} \right), \tag{3}$$

The first term on the right-hand side is based on volume conservation within a cylindrical cloud of constant $h$ in a non-divergent wind field. The second term accounts for outward spreading due to time-flattening of the cloud.

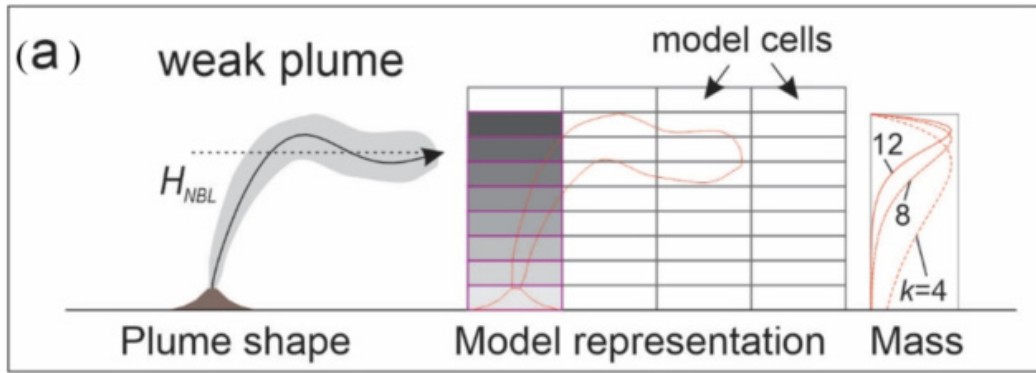

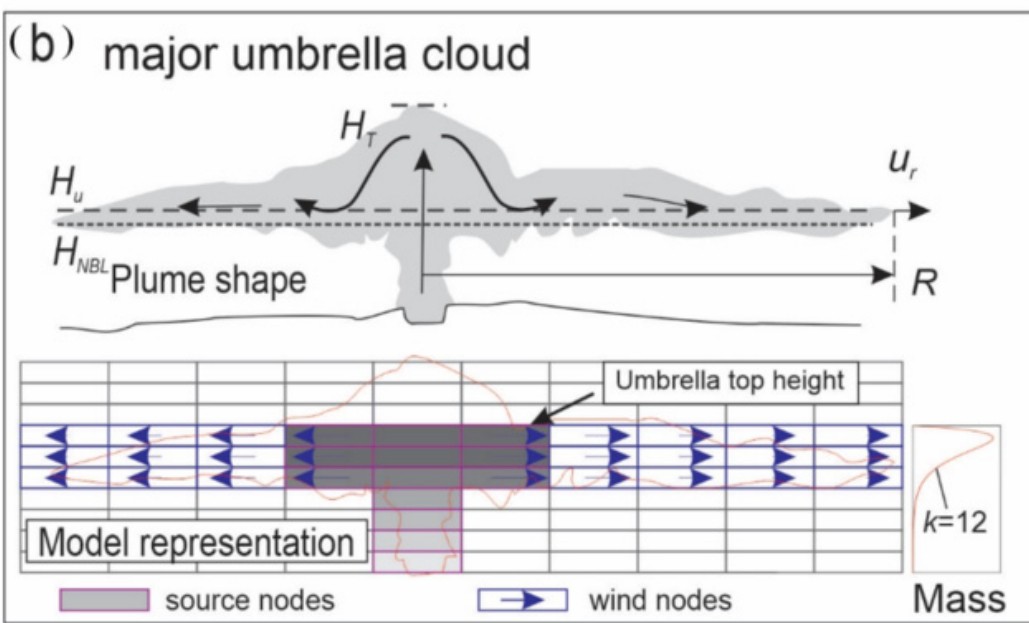

**Figure 1.** (**a**) illustration of the conventional configuration of source nodes in a Eulerian volcanic ash transport and dispersion (VATD) model. A weak bent plume (**left**) is represented as a column of source nodes (**center**) that lie vertically above the vent. The darkness of each source node in the figure represents the relative mass added to each node. Red curves to the right represent the vertical distribution of mass with height as specified using a Suzuki function with $k$ = 4, 8, and 12, respectively. (**b**) Depiction of the configuration of source nodes in the Ash3d model with an umbrella cloud. The source nodes in the model extend only to the top of the umbrella cloud (gray cells in lower figure). From the ground level to the base of the umbrella, the source nodes consist of a single column of nodes. Within the umbrella, the source nodes consist of a cluster, $3 \times 3$ nodes in map view, with enough nodes in the vertical dimension to extend from 75% to 100% of the umbrella-top height. A radial wind (blue arrows) is added to cells within the cloud, extending from the cloud center to its margins, with a wind speed specified by Equation (3) or (4). The vertical distribution of mass in the source nodes is given by a Suzuki function with $k$ = 12 (red curve at lower right).

Webster, et al. [2] derived an alternative formula by noting that $u_R \propto R^{-1/2}$ and extrapolating that relation to radial wind within the cloud:

$$u(r) = u_R \left(\frac{R}{r}\right)^{1/2}, \tag{4}$$

Both Equations (3) and (4) match the boundary condition that $u(r) \to u_R$ as $r \to R$; but Equation (3) gives significantly higher velocities when $r << R$. At a fixed point within the cloud, Equation (3) predicts an increase in $u(r)$ with time while Equation (4) predicts constant $u(r)$ [2]. Neither equation exactly conserves volume flux through the cloud. The different wind speeds predicted by these equations do not significantly affect the outline of the modeled cloud, but they do affect mass distribution within the cloud, and they could affect the mass distribution of a modeled deposit. We use Equation (3) to maintain consistency with earlier studies [4,18] but explore the different predictions of Equation (4) using the Calbuco deposit.

## 3. Model Implementation

We model ash transport using Ash3d, a Eulerian, finite volume code that calculates downwind advection, turbulent diffusion, and particle settling throughout the cells in a 3D atmospheric grid [19]. For operational runs that ignore an umbrella cloud we have used the column of nodes above the vent as source nodes (Figure 1a). We distribute mass among those cells using the Suzuki equation [20],

$$\frac{d\dot{M}}{dz_v} = \dot{M}\frac{k^2(1 - z_v/H_T)exp(k(z_v/H_T - 1))}{H_T[1 - (1 + k)exp(-k)]} \tag{5}$$

where $\dot{M}$ is the mass flux into the plume, $H_T$ is plume height above the vent, $z_v$ is the elevation above the vent within the plume, and $k$ is an adjustable constant that determines the distribution of mass. Distributions with k = 4, 8, and 12 are illustrated in Figure 1. For our simulations without an umbrella cloud we use $k = 8$, which was found in deposit simulations to fit well with observed deposit thickness versus distance [21].

To incorporate the growth of an umbrella cloud, we add ash to a cluster of cells centered over the vent, $3 \times 3$ in map view, at an elevation range extending from the umbrella top in Figure 1b to 75% of the umbrella top height. Ash is not placed into the overshooting top under the assumption that ash at these elevations collapses into the umbrella rather than being advected laterally (Figure 1b). The growth of the cloud radius $R$ with time $t$ is calculated using Equations (1) and (2).

Key model inputs include the eruption start time, duration, plume height, erupted mass or volume, and the distribution of particle size, shape, and density. The wind field is derived from 3D, time-dependent meteorological models. We use the ERA5 model from the European Centre for Medium-Range Weather Forecasts (ECMWF) [22]. This global model, released in 2019, has a horizontal resolution of 0.28°, time resolution of 1 h, and 137 hybrid sigma pressure levels extending to 0.01 hPa (~45 km altitude). It replaces the ECMWF ERA Interim model [23] (79 km horizontal resolution, 60 pressure levels to 0.1 hPa, and 6 h time steps). We have run simulations using both wind models and note that the ERA5 model produces a slightly better fit with observed cloud movement for our three eruptions (Figures S1–S3). To enable fast cloud simulations we assume a single size class of 0.01 mm diameter, density 2000 kg m$^{-3}$, which has a negligible settling velocity.

Next, we compare Ash3d simulations using the umbrella cloud formulation with observations from three well-documented eruptions.

## 4. Application to Eruptions

In recent decades, umbrella-cloud growth has been described at Mount St. Helens [14], Pinatubo [6], El Chichón [24], Kelud [8,16], Calbuco [9], Okmok, Sarychev, Grímsvötn, and Shishaldin [13], among others. Among those, we examine Pinatubo because of its widely studied umbrella growth, and Kelud

and Calbuco for their high-quality satellite observations. The deposit from Calbuco is also well mapped and its data are publicly available [9].

### 4.1. Pinatubo 15 June 1991

Pinatubo is the iconic eruption that demonstrated the effect of umbrella spreading. It was the largest in the world during the past several decades and the only VEI 6 + eruption to have occurred during the modern era of satellite observations. Observations of the massive growing cloud sparked many studies [6,17,25–27]. The eruption was also a landmark for aviation, producing more encounters with aircraft (17) than any prior to Eyjafjallajökull (91) [28,29].

The main event was preceded by two months of elevated seismicity, then four vertical eruptions on 12–14 June, and multiple surge-producing eruptions on 14–15 June [30]. Continuous, high-amplitude tremor starting at 05:42 UTC on 15 June signaled the onset of the main phase [31].

The cloud's massive size and the arrival of Typhoon Yunya in Luzon prevented ground-based visual observations, but hourly images by the Japanese Geostationary Meteorological Satellite (GMS or Himawari (4) and the U.S. National Oceanic and Atmospheric Administration's (NOAA's) polar-orbiting Advanced Very High Resolution Radiometer (AVHRR) satellite documented its evolution. Hourly outlines of the cloud (Figure 2a) based on GMS images published in Figure 5 of Holasek, et al. [7] show that it grew to 1220 km in diameter over 11 h (Figure 2a), although emission dropped after about 9 h [7]. The MER has been estimated by several methods at $10^8$–$10^9$ kg s$^{-1}$ [6,7,27,32]. Here, fitting a line of form $R = a(t - b)^{2/3}$ through the data (Figure 2b), and solving Equations (1) and (2) for the best-fit parameter $a$, we estimate MER $\approx 6.2 \times 10^8$ kg s$^{-1}$. The best-fit eruption start time of $b =$ 05:28 UTC is about fourteen minutes before the start of continuous tremor.

**Table 1.** Inputs to Ash3d simulations of Pinatubo.

| Parameter | Value |
|---|---|
| Eruption start (UTC) | 15 June 1991, 05:30 UTC * |
| vent latitude | 15.133° N |
| vent longitude | 120.350° E |
| Eruption duration (hrs) | 9:00 |
| max. plume height (km asl) | 35 † |
| erupted volume (km$^3$ DRE) | 8 * |
| erupted mass (kg) ** | $2 \times 10^{13}$ |
| umbrella cloud-top height (km asl) | 25 † |
| k (no umbrella) | 8 |
| k (with umbrella) | 12 |
| Diffusivity ($K$, m$^2$ s$^{-1}$) | 0 †† |
| horizontal nodal spacing (deg.) | 0.2 |
| vertical nodal spacing (km) | 2 |
| Grain size, shape, density‡ | single particle size, 0.01 mm, density = 2000 kg m$^{-3}$, $F = 0.44$ |
| C (m$^3$ kg$^{-3/4}$ s$^{-3/2}$) | 430 |
| N (s$^{-1}$) | 0.02 ‡‡ |
| λ | 0.2 ‡‡ |
| Meteorological model used | European Centre for Medium-range Weather Forecasts (ECMWF) ERA5 model |

Notes: * Start time of the main phase was derived from the best-fit curve of *R* vs. *t* in Figure 2b. Erupted volume was derived by taking the best-fit mass eruption rate (MER) in Figure 2b, multiplying by a 9-hour-long eruption, and using a magma density of 2500 kg m$^{-3}$ to convert from erupted mass to volume. ** Our simulations assume that only 5% of the erupted mass makes it into the cloud [33]. † Maximum plume height is used only in the simulation that does not include the umbrella cloud. Umbrella-top height is used in the Ash3d simulation that considers the umbrella cloud. These heights are based on estimates by Holasek et al. [7]. †† Diffusivity is set to zero to maximize the contrast between the umbrella and non-umbrella simulations. ‡ Fall velocity for all simulations is calculated using relations of Wilson and Huang [34], which approximates particles as ellipsoids having a shape factor $F \equiv (b + c)/2a$, where *a, b,* and *c* are the semimajor, intermediate, and semiminor axes of the ellipsoid. The value $F = 0.44$ is the average of shape factors of natural pyroclasts measured by Wilson and Huang [34]. A single particle size is used for cloud simulations to give rapid results. Settling velocity is negligible for particles of this size. ‡‡ λ = 0.2 and N = 0.017 s$^{-1}$ (rounded to 0.02) taken from Suzuki and Koyaguchi [17].

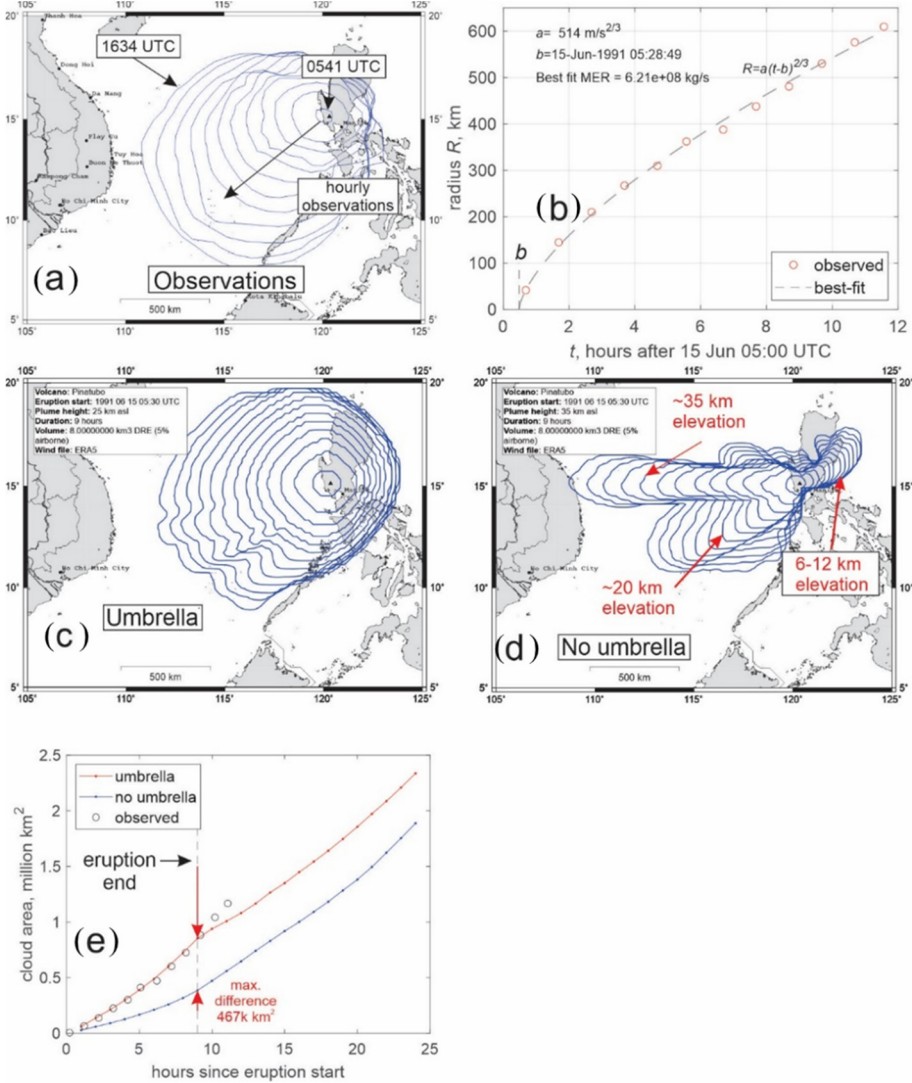

**Figure 2.** (**a**) Outlines of the growing Pinatubo umbrella on 15 June 1991 as digitized from images of the Geostationary Meteorological Satellite (GMS) satellite in Figure 5b of Holasek, et al. [7]. Blue lines give the outline of the cloud at 05:41, 06:41, 07:41, 08:41, 09:41, 10:34, 11:41, 12:41, 13:41, 14:41, 15:41, and 16:34 UTC. Due to the location of this satellite on the equator at 140° E longitude, boundaries of the airborne cloud are displaced northward and westward by 32% and 44% of their height, respectively [7]. Effects of image distortion and parallax on these outlines have not been analyzed. (**b**) Radius of the Pinatubo cloud versus time in hours after 15 June 05:00. Cloud radius ($R$) is calculated as $R = \sqrt{A/\pi}$, where $A$ is cloud area. The gray dashed line is a best-fit curve of form $R = a(t - b)^{2/3}$, where $a$ and $b$ are fitting coefficients having the values shown in the figure. (**c**) Ash3d simulation of the cloud at the same times as in Figure 2a, using inputs from Table 1 and calculating umbrella growth. The cloud margin is defined as the location where the cloud load equals 0.2 g m$^{-2}$. (**d**) Ash3d simulation using the inputs in Table 1, without an umbrella cloud. The location of Pinatubo is indicated by a triangle. (**e**) Cloud area versus time since the eruption start. The circles give the area of the cloud outlines shown in Figure 2a. The red line is the simulated area using the model that considers umbrella growth; the blue line represents the area simulated without considering umbrella growth.

The overshooting top-height during the main event was about 35–40 km above sea level (asl), or 34–39 km above vent level (avl) [7], but the umbrella cloud spread at altitudes of about 19–25 km asl [6,7]. We use these and other inputs (Table 1) to simulate the eruption both with the umbrella cloud (Figure 2c) and without (Figure 2d).

Several studies have had to modify the wind field in order to get the modeled Pinatubo cloud to match the movement of the observed cloud. Costa, et al. [3], Mastin, et al. [4], and Webster, et al. [2] for example had to rotate the wind field 30° counterclockwise and reduce wind speeds by up to a factor of two, using ERA Interim and the National Center for Atmospheric Research/National Center for Environmental Prediction (NCAR/NCEP) Reanalysis 1 wind fields. To produce Figure 2c using ERA5 winds, we were able to match the cloud movement direction with no rotation, but had to reduce wind speed by 30% to match travel distance (other variations, including the simulation with no wind modification, are illustrated in Figure S4). The cloud-movement direction was likely sensitive to height, as wind directions rotated strongly from N 60° E at 16.6 km to N 100° E at 22 km elevation (Figure 3a) [35]. Additional model simulations with an umbrella top-height dropped from 25 to 22.5, and 20 km, however, did not strongly change the visible direction of cloud movement (Figure S5).

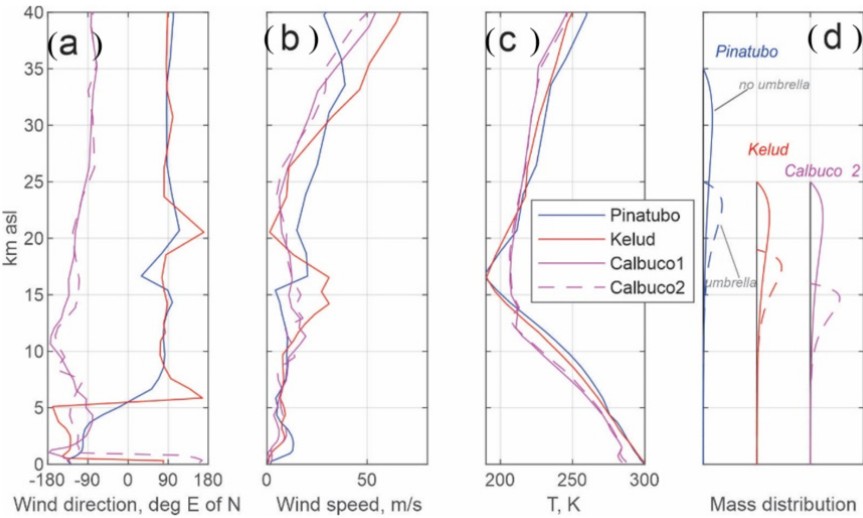

**Figure 3.** Wind direction (**a**), speed (**b**), and air temperature (**c**) as a function of altitude during the three eruptions studied. Vertical profiles for all eruptions are taken from the nearest node in the ERA5 data at times near the eruption start: 06:00 UTC for Pinatubo; 16:00 UTC for Kelud; 19:00 UTC on 22 April for Calbuco phase 1; and 06:00 UTC on 23 April 2015 for Calbuco phase 2. Wind direction is the direction from which the wind is blowing, in degrees clockwise from true north. "Calbuco1" and "Calbuco2" refer to phase 1 and 2 of the Calbuco eruption. (**d**) Vertical distribution of mass used in models of Pinatubo (**left, blue**), Kelud (**center, red**), and Calbuco phase 2 (**right, magenta**). Solid and dashed lines give mass distributions for the non-umbrella and umbrella simulations, respectively.

In our final simulations (Figure 2), the size and shape of the cloud is clearly more realistic when modeled with umbrella growth (Figure 2c), as previous studies have shown [2–4]. After 24 h, the simulated cloud area ($2.33 \times 10^6$ km$^2$, Figure 2e) is nearly $0.45 \times 10^6$ km$^2$ greater than the simulation would predict without the umbrella ($1.89 \times 10^6$ km$^2$), revealing the greater hazard. Over the first 9 h, the modeled cloud area (Figure 2e) tracks very closely with the observed area but diverges in the two hours thereafter. The divergence reflects flattening and spreading of the real cloud after the eruption subsided, which is not considered in our model. The divergence also reflects continued ash emission at a lower rate after 9 h, which cannot currently be simulated. Although the standard version of Ash3d allows users to input a time series of plume height and eruption rate, the umbrella-cloud version allows only a single pulse at a constant height and rate. We imposed this limitation based on our belief that sequential intrusions at multiple altitudes cannot yet be realistically modeled.

Some discrepancies between cloud movement in the umbrella and no-umbrella cases are surprising. For example, the no-umbrella case (Figure 2d) sends ash farther west than the umbrella case even though it lacks radial winds to boost it. The no-umbrella result also displays at least three lobes: one trending west, one southwest, and one northeast. Additional model outputs reveal that the longer

west-trending lobe travels at ~35 km asl; the southwest lobe at ~20 km, and the shortest northeast lobe at 6–12 km. For the no-umbrella simulation, we used the total 35-km plume height as input, and a *k* value (k = 8), which distributes ash more evenly from the base to the top of the column (Figure 3d). In the umbrella version (Figure 2c), source nodes extend only to the umbrella top; this fact, and larger *k*, concentrate most ash in a narrower vertical range and produce a cleaner umbrella shape. Whether these parameterizations improve the match with other umbrella clouds will only be seen by more comparisons.

## 4.2. Kelud, 13 February 2014

Kelud volcano is among the world's deadliest, with numerous Vulcanian eruptions that have thrown water out of its summit crater and generated lahars that swept down river valleys. Eruptions in 1586 and 1919 for example killed about 10,000 and 5,000 people, respectively [36]. Eight eruptions over the past century (1919, 1920, 1951, 1966, 1967, 1990, 2007–2008, and 2014) have killed over 6,000 people. Excavations into the crater wall to reduce the size of the crater lake have helped reduce fatalities but have not reduced the tendency for violent, Vulcanian outbursts.

Kelud eruption characteristics and model inputs are listed in Table 2. The 13 February 2014 eruption was preceded by about a week-long increase in seismicity and by a gradual increase in tilt starting in December 2013 [37]. The eruption onset was detected by a thermal infrared anomaly in the crater at ~15:46 UTC [16] and was associated with destruction of a lava dome and a lateral blast to the northeast [37]. The start of continuous acoustic tremor at 16:01:30 was interpreted by Nakashima, et al. [38] as the start of the Plinian phase. The eruption disrupted about 40 flights [39]; one flight from Perth, Australia to Jakarta, Indonesia, encountered ash when flying beneath the growing cloud [8]. The Plinian phase continued for about two hours, waning gradually from 18:00 to 18:30 [16,38,39]. Around 19:00–19:30, the plume had separated from the vent, marking the end of the eruption.

**Table 2.** Inputs to Ash3d simulations of Kelud.

| Parameter | Value |
| --- | --- |
| Eruption start (UTC) | 13 February 2014, 16:12 UTC * |
| vent longitude | 112.308° E |
| vent latitude | −7.903° N |
| Eruption duration (hrs) | 2:00 |
| max. plume height (km asl) | 25 ** |
| erupted volume (km$^3$ DRE) | 0.25 † |
| erupted mass (kg) | $6.3 \times 10^{11}$ |
| umbrella cloud-top height (km asl) | 19 ** |
| k (no umbrella) | 8 |
| k (with umbrella) | 12 |
| Diffusivity (K, m$^2$ s$^{-1}$) | 0 |
| horizontal nodal spacing (deg.) | 0.2 |
| vertical nodal spacing (km) | 2 |
| Total grain-size distribution | single particle size, 0.01 mm, density = 2000 kg m$^{-3}$, shape factor $F$ = 0.44 |
| C (m$^3$ kg$^{-3/4}$ s$^{-3/2}$) | 430 |
| $N$ (s$^{-1}$) | 0.02 †† |
| λ | 0.2 |
| Meteorological model used to provide the wind field | ECMWF ERA5 |

Notes: * Start time of main phase based on a regression through a plot of umbrella-cloud radius versus time. ** From space-based Light Detection and Ranging (lidar) measurements reported in Kristiansen et al. [8]. † Intermediate value between erupted volumes calculated from the best-fit MER values in Figure 4b. †† Calculated based on the method of Webster et al. [2], which uses only properties within the umbrella cloud. Hargie et al. [16] calculated $N$ = 0.015 based on atmospheric properties from the vent to the cloud top.

The growing umbrella could be tracked at 10-min intervals in infrared images of the Multi-functional Transport Satellite MTSAT-2 and MTSAT-1R geostationary platforms. The NASA/ Centre National d'Etudes Spatiales (NASA/CNES) satellite CALIPSO, with Cloud-Aerosol Lidar with Orthogonal Polarization (CALIOP) passed directly over the cloud at 18:12–18:14 UTC. A CALIOP curtain plot of total attenuated backscatter at 532 nm showed strong attenuation at about 18–19 km asl (17–18 avl) (Figure 2e of [8]), marking the top of the umbrella. Higher reflectors within the overshooting top could be seen to at least 26 km. Dispersion modeling by Kristiansen, et al. [8], using a Bayesian inverse method, estimated a maximum ash concentration of $9 \pm 3$ mg m$^{-3}$ encountered by the Perth-to-Jakarta aircraft. The same modeling results suggested that most of the ash was injected at elevations of 11, 15, and 27 km above sea level.

Hargie, et al. [16] compared observed rates of umbrella-cloud expansion with theoretical curves and showed an intensification of the eruption during the first hour, followed by steady growth for about an hour, and waning after about 17:50 UTC, with an average eruption rate between 16:01 and 18:00 UTC of $0.8–1.0 \times 10^8$ kg s$^{-1}$. Three-dimensional plume modeling by Suzuki and Iguchi [40] yielded an estimate of $0.3–1 \times 10^8$ kg s$^{-1}$.

On images from Japan's MTSAT-1R geostationary satellite, we have drawn two outlines on each image; one delineating the cloud tightly and the other more expansively (Figure 4a, Figure S6). Cloud areas are derived from these outlines. In Figure 4b, cloud radii (assuming $R = \sqrt{A/\pi}$), are fit to two curves of form $R = a(t - b)^{2/3}$. The best -fit values of $a$ are used with Equations (1) and (2) to estimate MER values of 0.58 and $1.3 \times 10^8$ kg s$^{-1}$—within the range of previous estimates. Best-fit eruption start-times (b) of 16:12 and 16:13 UTC are also similar to Hargie et al.'s (16:09). Using a 2-hour duration and average magma density of 2500 kg m$^{-3}$, these MER values translate to erupted volumes of 0.17 and 0.37 km$^3$ DRE respectively.

For our simulations we use an intermediate volume of 0.25 km$^3$ DRE and other inputs in Table 2. The cloud produced by the umbrella simulation (Figure 4c) is clearly closer in size and shape to the observed cloud than that of the non-umbrella simulation (Figure 4d). The wind speeds at the umbrella height (Figure 3b) are among the highest in the atmosphere between the vent and $H_T$, producing an umbrella cloud that (unlike Pinatubo) is not outpaced by any lobes in the non-umbrella simulation. In the umbrella simulation, the cloud area grows with time at a rate closely following the observed rate using the larger outlines (Figure 4e,f); but unlike at Pinatubo, the growth rate of the observed cloud tails off rapidly after about two hours, while the modeled cloud continues to enlarge. The no-umbrella simulation grows more slowly and is about 38,000 km$^2$ smaller after four hours. However, over the next 12 h, the non-umbrella cloud area converges with and eventually surpasses that from the umbrella simulation.

The divergence in area of the modeled cloud from the observed cloud after the eruption end is surprising. A simple explanation may be that advection is spreading the cloud after the eruption stops, but it is unclear why this would spread the modeled cloud more than the observed cloud. Some of the expansion may be taking place at lower elevations and warmer infrared temperatures that are not apparent in the infrared images.

### 4.3. Calbuco, 22–23 April 2015

The 2015 eruption of Calbuco began on 22 April at 21:04 UTC, after only a few hours of precursory seismicity. Chile's Servicio Nacional de Geología y Minería (SERNAGEOMIN) reported a 90-min gray ash plume that drifted mainly ENE, although fine ash drifted N and NW. A second pulse began seismically at about 04:00 UTC on 23 April and continued for about six hours. Plume heights were reported by SERNAGEOMIN to be above 15 km asl (13 km avl) [41], although later, polarimetric radar analysis suggested maximum heights above 25 km for both phases [42]. Sporadic activity continued for the next several days, though no plumes rose more than a few kilometers above the summit. Tephra fall damaged structures [43], impaired road transportation [44], and caused flight cancellations for several days.

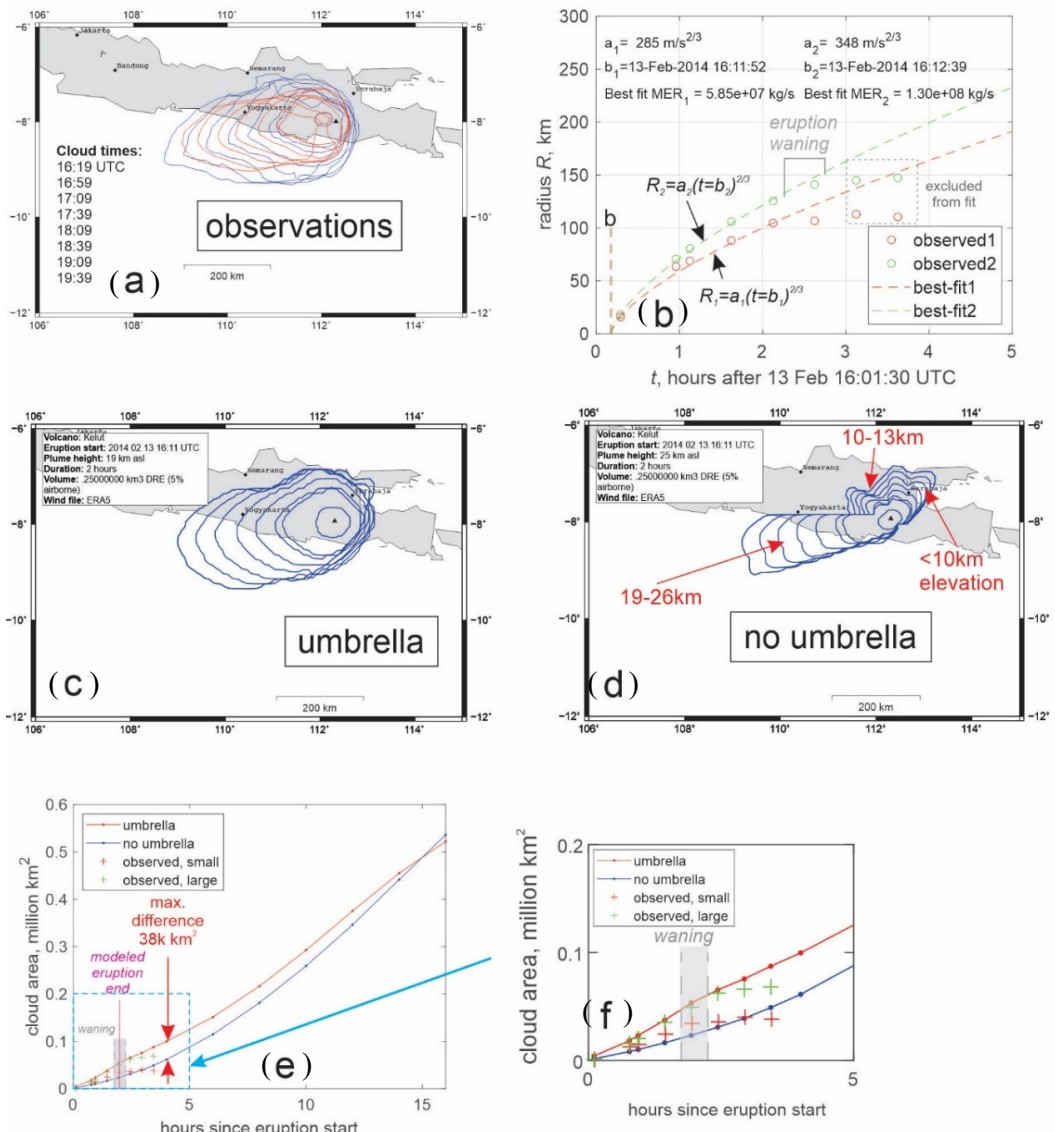

**Figure 4.** (**a**) Outlines of the Kelud cloud on 13 February 2014 as digitized in MTSAT-1R IR channel 13 images. The exact footprint was frequently ambiguous in these grayscale IR images, so we digitized two cloud outlines, one having a relatively smaller and the other a larger footprint. Blue lines illustrate the larger footprint, red lines the smaller. Satellite images with the hand-drawn lines superimposed are shown in Figure S6. The outlines correspond (from inner to outer) to times of 16:19, 16:59, 17:09, 17:39, 18:09, 18:39, 19:09, and 19:39 UTC. Due to the location of the satellite, 140° E longitude on the equatorial plane, parallax displaced clouds at 20 km altitude about 9 km WNW in map view. Effects of image distortion and parallax on these outlines have not been analyzed. (**b**) Radius of the Kelud cloud versus time in hours after 13 February 16:01:30 UTC. Cloud radius ($R$) is calculated as $R = \sqrt{A/\pi}$, where $A$ is cloud area. Green and red dashed lines are best-fit curves of form $R = a(t-b)^{2/3}$ through the green and red dots, respectively, where $a$ and $b$ are fitting coefficients having the values shown in the figure. (**c**) Ash3d simulation of the cloud, using inputs from Table 2 and calculating umbrella growth. Blue lines are cloud outlines at the same times as in Figure 2a. The cloud margin is defined as the location where the maximum cloud mass load equals 0.2 g m$^{-2}$. The location of Kelud is marked by the black triangle. (**d**) Ash3d simulation using the inputs in Table 2, without an umbrella cloud. (**e**) Cloud area versus hours after the eruption start. Crosses are observations from Figure 4b; the red line is the model forecast including umbrella growth, and the blue line is the model forecast not including umbrella growth. (**f**) Close-up of the first five hours of cloud growth shown in Figure 4e.

Eruption characteristics and model inputs are summarized in Table 3. Growth of the umbrella cloud from phase 1 was visible to ground observers and to the Geostationary Operational Environmental Satellite-13 (GOES-13), which returned images roughly every half hour. From the satellite images, Romero, et al. [10] noted that the travel direction of the phase 1 cloud was N48° E while that of the phase 2 cloud was N55° E. Six hours after the phase 2 eruption started, the cloud in GOES images displayed a NE-trending fan shape with its apex near the volcano, spanning the entire NE quadrant from N0° E to N90° E.

**Table 3.** Inputs to Calbuco model simulations. Eruption start times and durations are taken from observations reported in Van Eaton, et al. [9].

|  | Phase 1 | Phase 2 |
|---|---|---|
| Eruption start (UTC) | 22 April, 21:04 | 23 April, 04:01 |
| vent latitude | −41.326N | same |
| vent longitude | −72.614E | same |
| Eruption duration | 1:31 | 6:14 |
| max. plume height (km asl) | 25 | 25 |
| erupted volume (km$^3$ DRE) | 0.033 * | 0.12 * |
| erupted mass (kg) | $8 \times 10^{10}$ | $3 \times 10^{11}$ |
| umbrella cloud height (km asl) | 16 | 16 |
| $k$ (no umbrella) | 8 | 8 |
| $k$ (with umbrella) | 12 | 12 |
| Diffusivity ($K$, m$^2$ s$^{-1}$) | 0 | 0 |
| horizontal nodal spacing (deg.) | 0.05 | 0.05 |
| vertical nodal spacing (km) | 2 | 2 |
| TGSD for cloud simulations | single particle 0.01 mm, density = 2000 kg m$^{-3}$ shape factor $F = 0.44$ | same |
| TGSD for deposit simulations | see Table 4 † | see Table 4 † |
| $C$ (m$^3$ kg$^{-3/4}$ s$^{-3/2}$) | 870 | 870 |
| $N$ (s$^{-1}$) | 0.02 ** | 0.02 ** |
| $\lambda$ | 0.1 | 0.1 |
| Meteorological model used to provide the wind field | ECMWF ERA5 | ECMWF ERA5 |

Notes: * The DRE erupted volume for phase 2 is an approximate average of the values that would be obtained using the average of the mass eruption rates estimated in Figure 5b, assuming an eruption duration of 6:14, and an average magma density of 2500 kg m$^{-3}$. The volume of phase 1 was calculated using the MER of Van Eaton et al. [9], but changed from $6 \times 10^6$ to $1.5 \times 10^7$ kg s$^{-1}$ by using $\lambda = 0.1$ in the calculation, which is recommended for mid-latitude eruptions by Suzuki and Koyaguchi [17]. ** Brunt-Väisälä frequency calculated using the formula in Equation (10) of Mastin [45], but integrating from the base to the top of the umbrella cloud (12–16 km asl) rather than from the vent elevation to the plume top. † We use different total grain-size distributions (TGSDs) for the cloud and the deposit because the type of output specified is different. Cloud simulations run faster when a single grain size is used, and coarser sizes are unnecessary.

Using the GOES-13 10.7 µm-wavelength thermal infrared channel, Van Eaton, et al. [9] outlined the cloud within each image, defined by areas within the −3 °C and −13 °C isotherms, respectively. Areas within each outline were calculated, as were equivalent radii. By plotting cloud radius versus time and comparing with theoretical curves predicted by Equations (1) and (A1), Van Eaton, et al. [9] estimated the mass eruption rate to be $6 \times 10^6$ kg s$^{-1}$ for phase 1, and $6.6$–$9.5 \times 10^6$ kg s$^{-1}$ for phase 2, implying a total erupted mass of $2 \times 10^{11}$ kg, which is almost within the mass uncertainty of the deposit ($4.5 \pm 2.2 \times 10^{11}$ kg). This MER is about an order of magnitude lower than that inferred from $H_T = 23$ km avl (25 km asl) using the relation of Mastin, et al. [32]. Van Eaton, et al. [9] also note that the phase 2 cloud stopped growing between 05:38 and 06:38 UTC although the eruption continued. The end of umbrella growth coincided with an abrupt increase in lightning activity. Both were inferred

to be related to the onset of pyroclastic density currents which may have consumed a significant fraction of erupted mass.

**Table 4.** Total grain-size distribution (TGSD) used in Calbuco deposit simulations. Rows in italics represent aggregates of fine ash. The TGSD is based on a simplification of the Mount St. Helens 1980 total grain-size distribution [21]. Particle sizes larger than 2 mm are consolidated into the 2 mm bin to speed calculation. The size distribution of aggregates was derived by optimizing the fit between model simulations and four well-mapped deposits [21]. Densities of each size fraction are based on densities of those component sizes at Mount St. Helens. Ash3d uses a fall velocity calculation based on Wilson and Huang [34] in which non-spherical particles are represented as ellipsoids with a shape factor $F = (b + c)/2a$, where $a$, $b$, and $c$, are the semi-major, intermediate, and semi-minor axes of the ellipsoid. A shape factor of 0.44 is the average of values of real pyroclasts measured by Wilson and Huang [34].

| Diameter mm | Mass Fraction | Density kg m$^{-3}$ | Shape Factor $F$ |
|---|---|---|---|
| 2 | 0.0611 | 800 | 0.44 |
| 1 | 0.07098 | 1040 | 0.44 |
| 0.5 | 0.22701 | 1280 | 0.44 |
| 0.25 | 0.21868 | 1520 | 0.44 |
| 0.1768 | 0.05326 | 1640 | 0.44 |
| 0.125 | 0.04039 | 1760 | 0.44 |
| 0.088 | 0.02814 | 1880 | 0.44 |
| *0.2176* | *0.018* | *600* | *1.0* |
| *0.2031* | *0.072* | *600* | *1.0* |
| *0.1895* | *0.12* | *600* | *1.0* |
| *0.1768* | *0.072* | *600* | *1.0* |
| *0.1649* | *0.018* | *600* | *1.0* |

For phase 2, we have also outlined the cloud in GOES-13 10.7 µm satellite images (Figure S7). Our outlines are based not on temperature thresholds but visual gradients in the grayscale images that might suggest cloud edges. In each image we have drawn two outlines, one smaller and the other larger, in an attempt to account for uncertainty. The larger outlines are illustrated in Figure 5a. The equivalent radii $R$ (assuming $R = \sqrt{A/\pi}$) are plotted as a function of time in Figure 5b. Best-fit lines through these points, in combination with Equations (1) and (2), yield MER = 0.9 and $1.6 \times 10^7$ kg s$^{-1}$, respectively. The average of these rates, multiplied by the eruption duration, yields a DRE volume of about 0.12 km$^3$ assuming a magma density of 2500 kg m$^{-3}$. Our cloud-based estimates of erupted volume are larger than those of Van Eaton, et al. [9] primarily because we used $\lambda = 0.1$ (following Suzuki and Koyaguchi [17]) whereas Van Eaton, et al. [9] used $\lambda = 0.2$ (following Costa, et al. [3]). Our larger estimate is closer to published volumes based on deposit mapping [9,10,46].

Simulations using this erupted volume, a 6.25-hour duration, and other parameters in Table 3, yield a cloud evolution as shown in Figure 5c with the umbrella, and 5d without. Clouds produced by umbrella simulation are more roundish than the non-umbrella outlines and cover an area larger than both the non-umbrella simulation and the observed cloud. By the end of the eruption, the cloud area in the umbrella simulation is about 36,000 km$^2$ greater than the non-umbrella area (Figure 5e). The discrepancy diminishes with time. Twenty-four hours after the eruption start, the cloud area from the umbrella simulation is slightly smaller than that from the non-umbrella simulation. The importance of wind in ash dispersal can also be seen by cloud displacement; by the eruption end at about 10:30 UTC, the cloud's southwest edge had been displaced about 100 km NE from the SW edge of the circular footprint that would have existed in the absence of wind (Figure 5c).

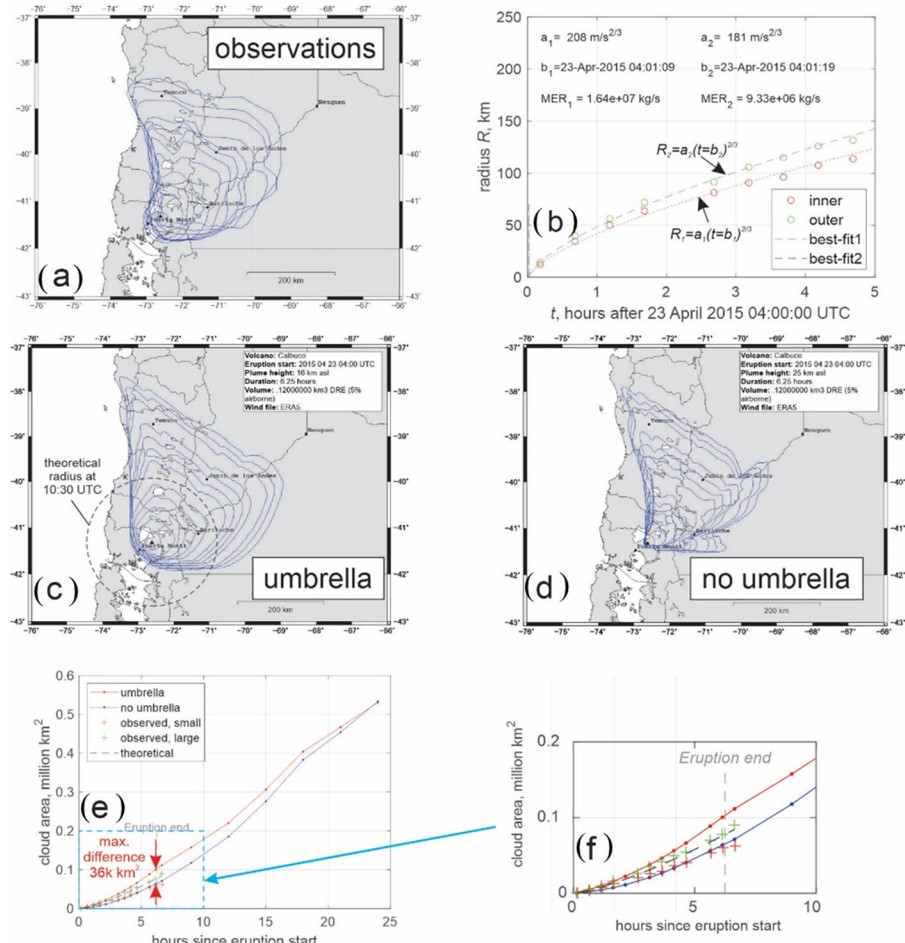

**Figure 5.** (**a**–**f**): plots analogous to those of Figure 4a–f, but for phase 2 of the Calbuco eruption. The blue cloud outlines in (**a,c,d**), were taken from images at 04:08, 04:38, 05:08, 05:38, 06:38, 07:08, 07:38, 08:08, 08:38, 09:38, 10:08, and 10:38 UTC on 23 April 2015. In (**a**), only the larger cloud outlines are shown.

The modeled cloud area after 6 h is also about 25,000 km² larger than the theoretical cloud area (black dashed line, Figure 5e,f). We have found that this discrepancy varies somewhat with model resolution (Figures S8 and S9), with a resolution of about 0.05° or less required for stable results. A resolution of 0.05° has an average cell dimension of about 5 km, roughly 3% of the 157-km umbrella radius at the end of the eruption (Figure 5c). Overall, the cloud produced by the non-umbrella simulation appears more realistic in both size and shape to the observed cloud. This result contrasts with those from the larger eruptions at Kelud and Pinatubo, where results of the umbrella simulation appeared more realistic. This result may suggest a lower limit to the size and/or wind conditions under which umbrella simulations are warranted.

Calbuco Deposit

Published deposit data for the Calbuco eruption gives us the opportunity to compare results with model simulations. Van Eaton, et al. [9] recorded sample thicknesses and stratigraphy at 163 sites out to ~460 km downwind. Their dispersal axis trends about N41° E toward Junin de los Andes, then turns ENE toward Neuquen, 480 km distant at a bearing of N57° E (Figure 6a). Romero, et al. [10] collected field data on thickness and maximum clast size at another 65 sites. Their isopach map resembled that of Van Eaton, et al. [9] but also included a lobe of <1 mm thickness that extended nearly 650 km northward, beyond Concepción on the west coast (Figure 6a). Integration of their tephra volume using exponential

thinning, power law, and Weibull methods yielded DRE volumes of 0.11–0.13 km$^3$. Castruccio, et al. [46] also mapped the deposit and estimated its volume as 0.28–0.38 km$^3$ bulk (0.11–0.15 km$^3$ DRE). Similar integration by Van Eaton, et al. [9] yielded 0.18 ± 0.09 km$^3$. Romero, et al. [10] were able to identify four main stratigraphic units in the deposit; the lower two trended in a more northerly direction and were consistent with the direction of movement of the phase 1 umbrella cloud. Between 22 April 21:00 UTC and 23 April 06:00 UTC, the wind directions at 9–15 km elevation rotated clockwise as much as 29° (Figure 3a), resulting in slightly more east-trending cloud movement in phase 2 [10].

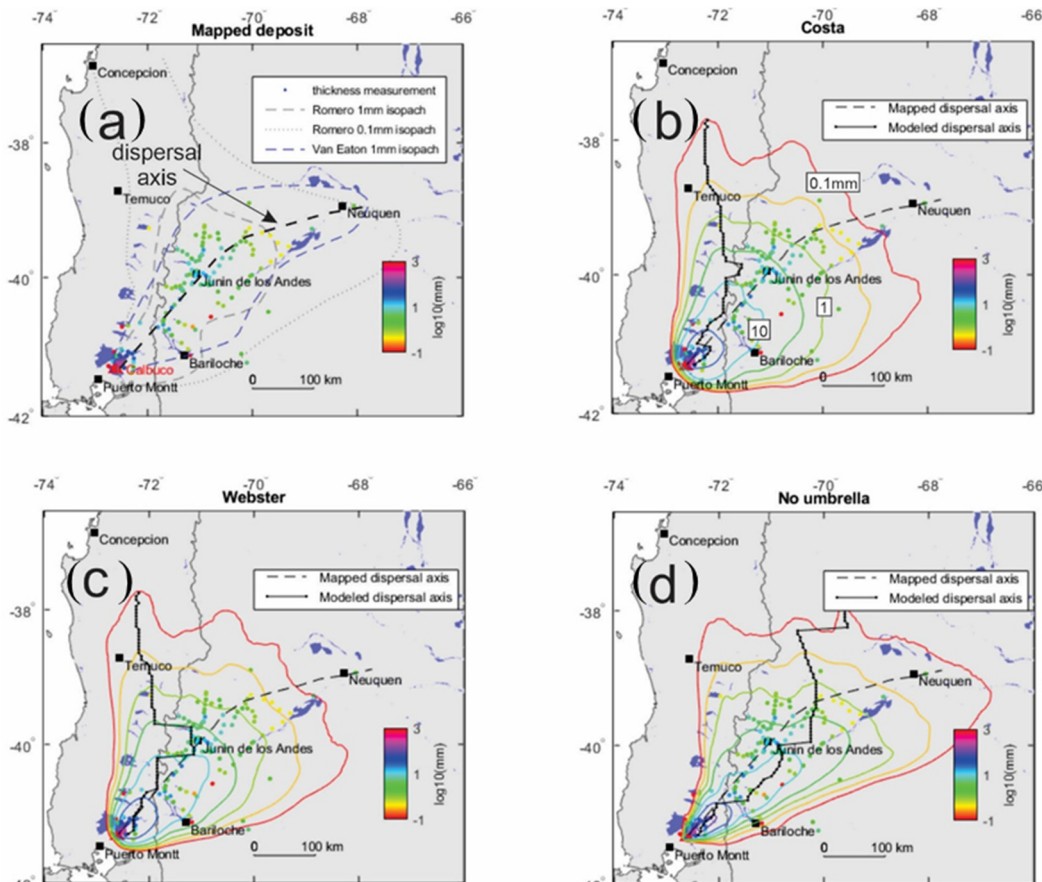

**Figure 6.** (**a**) extent of deposits from the 2015 Calbuco eruption, mapped by Van Eaton, et al. [9] and Romero, et al. [10]. Blue dashed line is the 1 mm isopach line mapped by Van Eaton et al.; gray dashed line is the 1 mm isopach mapped by Romero et al.; gray dotted line is the 0.1 mm isopach mapped by Romero et al. Colored dots are mean values of a thickness range at each location given by Van Eaton, et al. [9]. Colors correspond to thickness as indicated in the color bar on the right. (**b**) Modeled thickness isopachs (colored lines) resulting from Ash3d simulation considering growth of an umbrella cloud and radial winds within the cloud as given in Equation (3). All other inputs are as given in Table 3. Thickness values assume a deposit density of 800 kg m$^{-3}$. Isopach colors correspond to thickness as given on the color bar, with values of 0.1, 0.32, 1, 3.2, 10, and 32 mm from outer to inner. Colored dots show measured sample thicknesses, using the same color scale as the isopachs. Black dashed line is the dispersal axis mapped by Van Eaton, et al. [9]. Black solid line is the dispersal axis of the modeled deposit, determined by finding the cell in each row of cells with the greatest deposit thickness, and drawing a line connecting those cells. (**c**) Thickness isopachs derived from an Ash3d simulation that considers growth of an umbrella cloud and radial winds as given in Equation (4). (**d**) Thickness isopachs resulting from an Ash3d simulation that does not include umbrella growth. Height used in (**b**,**c**) is that of the umbrella-top—16 km asl. Height used in (**d**) is the plume top—25 km asl.

Our umbrella source module in Ash3d does not allow for simulation of a multi-pulse eruption. To simulate the deposit, we combined deposits from two single-pulse events. Inputs for those events are given in Table 3. We used a total grain-size distribution (TGSD, Table 4) that is a simplification of the Mount St. Helens 1980 TGSD, with fine ash represented by aggregates having a lognormal size distribution that has been optimized to match the deposits of four well-mapped eruptions [21]. (Cloud simulations in this paper use a single particle size, 0.01mm, with negligible settling velocity, representing a passive tracer.) We simulated umbrella growth using radial wind speeds calculated using both Equation (3) by Costa, et al. [3] (Figure 6b), and Equation (4) by Webster, et al. [2] (Figure 6c). We also simulated the deposit using the classical formulation that ignores umbrella growth (Figure 6d).

The dispersal pattern for these three cases is broadly similar; the deposit for the Costa-wind umbrella case (Figure 6b), which uses the strong radial wind field of Equation (3), is a bit more circular but otherwise similar to that predicted (Figure 6c) using the weaker radial wind field of Equation (4). Moreover, the southern boundary of the deposit extends a bit farther south in both umbrella simulations (Figure 6b–c) than in the non-umbrella simulation (Figure 6d).

The model isopachs in Figure 6b–d reasonably outline the mapped deposit as indicated by the cluster of sample locations. The sample thicknesses (as indicated by colors of dots) also agrees reasonably well with model thicknesses (given by colors of lines), especially for the yellow (0.3 mm) and green (1 mm) dots along the dispersal axis. Turquoise (10 mm) dots extend a bit farther down the dispersal axis than the modeled isopach lines. The correlation degrades at more distal regions. In particular, the 0.1 mm isopach of Romero, et al. [10] and the 1 mm isopach of Van Eaton, et al. [9] both extend much farther than any modeled isopachs.

This discrepancy in the extent of the 0.1mm isopach may result from at least two issues. First, to convert Ash3d calculations of deposit mass per unit area (mpua) to thickness, we use a density of 800 kg m$^{-3}$, consistent with previous studies [9,10]. Site-by-site deposit densities at Calbuco have not been published; but at some other deposits, e.g., Mount St. Helens in 1980, they decreased systematically from >800 to <100 kg m$^{-3}$ with distance [47]. If a similar trend exists at Calbuco, assuming a constant density in modeling could result in underestimates of modeled thicknesses by up to a factor of 10 at distal locations. Secondly, our choice of TGSD, which represents all fine ash as aggregates having a narrow size range of 0.165–0.218 mm (Table 4), was derived from a study [21] that adjusted aggregate size to optimize the agreement between modeled and mapped deposits at locations where most fine ash fell, usually at a secondary thickness maximum. The aggregate size distribution was not adjusted to optimize the fit to mapped deposits in very distal locations. Adding smaller or less dense aggregates would produce a distal tail to the deposit that extends farther downwind. We have done some simulations using an aggregate size distribution that is skewed with a fine tail (Figure S10) and found a slight qualitative improvement in fit, but more work is needed.

Optimizing the fit between the modeled and mapped deposits would require many simulations that systematically vary source parameters, which is beyond the scope of this work. Even the best efforts to match modeled deposits with mapped ones include many locations where discrepancies in thickness exceed a half order of magnitude [21,48]. All three of these simulations provide reasonable approximations of the areal distribution of ash and its thickness. None is clearly better than any other.

## 5. Discussion

We know that umbrella-cloud growth disperses volcanic ash by radial winds that are not considered in conventional advection–dispersion models. We have now studied three umbrella-forming eruptions of different sizes to see how much umbrella spreading affects ash dispersal. The results are summarized in Table 5. In our simulation, the VEI 6 Pinatubo eruption's 1200-km-diameter cloud covered more than twice the area predicted by the no-umbrella model (Figure 2e) and the greater area persisted throughout the 24-hour simulation. The smaller, mid-VEI 4 Kelud eruption produced a cloud that covered about 50,000 km$^2$ by the end of the eruption and was also more than twice the area predicted by the no-umbrella model (Figure 4e). However, that discrepancy diminished with time and actually

became negative (i.e., the cloud in the no-umbrella simulation was larger than that in the umbrella simulation) after about 15 h. In the smaller VEI Calbuco phase 2 eruption, the discrepancy also diminished with time and became about 3000 km$^2$ negative after about 24 h (Figure 5e).

**Table 5.** Summary of results for Pinatubo, Kelud and Calbuco. Mass eruption rate (MER) is based on best-fit curves in Figure 2b, Figure 4b, and Figure 5b. Volume is based on considerations given in the text. The column headings "Obs", "Umb", and "None", give the areas of clouds observed (Obs), predicted by models that consider umbrella growth (Umb), and by models that do not consider umbrella growth (None). The "performance" row assesses qualitatively whether simulations that consider umbrella growth are better or worse at reproducing the size and shape of the observed cloud. "N/A" indicates that cloud-area data were not analyzed at the times indicated. "N/A" indicates that cloud area observations were not analyzed at the times indicated.

| | **Pinatubo** | | | **Kelud** | | | **Calbuco Phase 2** | | |
|---|---|---|---|---|---|---|---|---|---|
| MER | ~6 × 10$^8$ kg s$^{-1}$ | | | 6–13 × 10$^7$ kg s$^{-1}$ | | | 9–16 × 10$^7$ kg s$^{-1}$ | | |
| Volume | ~8 km$^3$ DRE | | | ~0.25 km$^3$ DRE | | | ~0.15 km$^3$ DRE | | |
| | **Cloud Areas, thousand km$^2$** | | | | | | | | |
| | **Umb** | **None** | **Obs** | **Umb** | **None** | **Obs** | **Umb** | **None** | **Obs** |
| Eruption end | 851 | 385 | 883 | 53.2 | 23.3 | 34–49 | 100 | 64 | 60–77 |
| After 24 h | 2336 | 1887 | N/A | 565 | 589 | N/A | 530 | 533 | N/A |
| Performance | better | | | slightly better | | | slightly worse | | |

We previously assumed that the inclusion of umbrella spreading in model simulations would result in larger clouds. Our results lead to the surprising conclusion that, under some circumstances, including umbrella growth may actually reduce cloud area, if that inclusion also reduces the vertical extent of ash. Whether cloud area increases or decreases depends on the amount of wind shear in the atmosphere, and the distribution of mass with height in the plume. Mass height distribution (MHD) is not well constrained. Our best hope for understanding how mass is distributed with height in plumes may come from model inversion [49–51]. However, such studies are few, and future progress will depend on the occurrence of new eruptions with good coverage from modern, high-resolution, geostationary multi-band satellites and other well-constrained eruption source parameters (ESPs).

In the meantime, operational models must assume some mass height distribution, however crude. Most operational models assume either that the mass distribution is uniform or that it follows a Suzuki curve from the vent to $H_T$ [52]. However, models that consider umbrella spreading must transition smoothly from small eruptions, where mass extends up to $H_T$, to large umbrella-forming ones, where mass extends up to approximately the umbrella top. Methods for this transition are experimental, but two possibilities are illustrated in Figure 7. Alternatively, an operational model could set a threshold eruption size, above which umbrella growth would automatically be considered. A threshold of 0.1–0.3 km$^3$ DRE might be appropriate based on these results (Table 5).

Good data on the deposits from umbrella-forming eruptions are even harder to find than data on the clouds. Our Calbuco simulations using both the non-umbrella simulation and two umbrella versions, agreed qualitatively with the mapped deposit distribution. Our confidence in modeling deposits from umbrella-forming eruptions will come with more data, which will become available only after more large, well-studied eruptions.

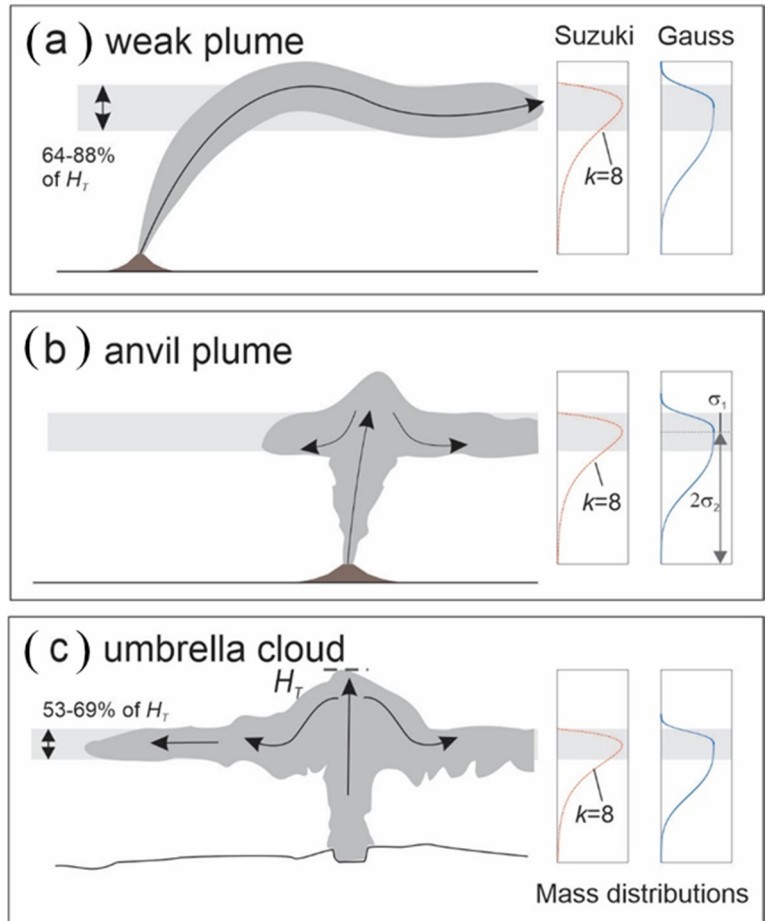

**Figure 7.** Two possible schemes for assigning vertical mass distribution to plumes of different size. Illustrated is (**a**) a weak plume, (**b**) an anvil plume, and (**c**) an umbrella-forming plume. Gray horizontal region in (**a**) represents the mean + /- standard deviation of neutral buoyancy heights for a weak, wind-driven plume calculated by one-dimensional plume models presented in Table 8 of Costa, et al. [53]. Gray bar in (**c**) represents the same range obtained by 1-D models for strong plumes (Table 12 of [53]). The gray bar in (**b**) represents a height range intermediate between (**a**) and (**c**). The cloud mass is assumed to be concentrated within these height ranges. On the right-hand side are possible vertical mass distributions with height. The red curves are Suzuki distributions with $k = 8$, extending from the vent to the top of the gray bar. Blue curves are an asymmetrical Gaussian distribution, with a maximum value in the center of the gray bar. Above that point, the standard error ($\sigma_1$) is the bar's half-height. Below the center, the standard error ($\sigma_2$) is half the distance to the vent elevation.

## 6. Conclusions

The largest volcanic eruptions can affect the broadest geographic areas and present the greatest hazard to both aviation and ground-based communities. Such eruptions produce their own wind field through umbrella spreading. Forecasting areas impacted by such eruptions require models that consider the umbrella-spreading process. In this paper, we have compared results of Ash3d simulations that include umbrella spreading with observation from three clouds and one deposit. The results, for the first hours of the eruption, offer a better visual agreement with the observed cloud than simulations that do not consider umbrella growth. We also find, surprisingly, that umbrella-cloud simulations do not always predict a larger cloud than simulations that don't consider umbrella spreading. The coming era of high-resolution satellite data may offer more observations that can improve model performance.

**Supplementary Materials:** The following are available online at http://www.mdpi.com/2073-4433/11/10/1038/s1, Figure S1: Effect of wind model used on the direction of cloud movement in the Pinatubo eruption. Figure S2: Effect of meteorological model on simulation results for the Kelud eruption. Figure S3: Effect of meteorological model on simulation results for the Calbuco eruption. Figure S4: Effect of modifications to the wind field on the Pinatubo cloud growth pattern. Figure S5: Effect of Pinatubo umbrella cloud-top height on the direction of ash dispersal. Figure S6: Channel-7 brightness temperature of the Kelud cloud from MTSAT-1R images. Figure S7: Channel-7 brightness temperature of the Calbuco phase 2 cloud from GOES-13 images. Figure S8: Calbuco cloud area versus time at four different horizontal model resolutions. Figure S9: Outline of the Calbuco phase 2 modeled cloud and the observed cloud at 1038 UTC. Figure S10: Effects of a fine tail in the grain-size distribution on the model deposit at Calbuco.

**Author Contributions:** Formal analysis, L.G.M.; Investigation, A.R.V.E. & L.G.M.; Methodology, A.R.V.E.; Writing—original draft, L.G.M. All authors have read and agreed to the published version of the manuscript.

**Funding:** Funding for this research was provided by the U.S. Geological Survey, Volcano Hazards Program.

**Acknowledgments:** This study has been conducted under the U.S. Geological Survey's Volcano Hazards program. We benefitted from extensive discussions with Helen Webster about the dynamics of umbrella-cloud development, and from a thorough manuscript review by Hans Schwaiger. We also thank three anonymous journal reviewers, one of whom recommended that we change $\lambda$ from 0.2 to 0.1 when estimating MER for Calbuco. Changing $\lambda$ to 0.1 increased the MER by a factor of two, putting our cloud-based estimates of erupted volume into better agreement with those from deposit mapping.

**Conflicts of Interest:** The authors declare no conflict of interest in this work.

**Appendix A**

The formula for Q originally published by Costa, et al. [3] for volume flux into the umbrella cloud was

$$Q = C\sqrt{k}\frac{\dot{M}^{3/4}}{N^{5/8}} \tag{A1}$$

Values of C were given as $0.5 \times 10^4$ m$^3$ kg$^{-3/4}$ s$^{-7/8}$ for tropical eruptions and $1.0 \times 10^4$ m$^3$ kg$^{-3/4}$ s$^{-7/8}$ for mid-latitude and polar eruptions. This formula was used in the study that introduced the umbrella-cloud implementation in the Ash3d model [4], and by Van Eaton, et al. [9] to estimate mass eruption rate from umbrella-cloud growth in the Calbuco eruption. Using the re-calibrated values of *C*, the corrected formula, given in an erratum to Costa, et al. [3], gives essentially the same values of *Q* as the old formula. For example, assuming $\dot{M} = 7.5 \times 10^6$ kg s$^{-1}$, $k_e = 0.1$, and $N = 0.02$ s$^{-1}$, the original formula gives $Q = 5.226 \times 10^9$ m$^3$ s$^{-1}$ whereas the new formula gives $Q = 5.242 \times 10^9$ m$^3$ s$^{-1}$.

Data Resource: Mastin, L.G., Van Eaton, A.R., 2020, Observations and model simulations of umbrella-cloud growth during eruptions of Mount Pinatubo (Philippines, 15 June 1991), Kelud Volcano (Indonesia, 14 February 2014), and Calbuco Volcano (Chile, 22–23 April 2015): U.S. Geological Survey data release, https://doi.org/10.5066/P9NPYCRH.

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
