# Peer review of "Comparing Simulations of Umbrella-Cloud Growth and Ash Transport with Observations from Pinatubo, Kelud, and Calbuco Volcanoes"

_atmosphere, doi:10.3390/atmos11101038_

Round 1
Reviewer 1 Report
Dear Editor,
This paper presents results from modelling experiments using an atmospheric dispersion model to investigate the impact of parameterising umbrella spreading on airborne volcanic ash cloud extent and deposit characteristics. Results are used to compare, and validate, parameterisations of umbrella cloud spreading. The paper and addresses a topic which is well known from observations of large eruptions but does not always receive sufficient attention from dispersion modellers. However, as the paper shows, umbrella cloud spreading can have a significant impact on modelled volcanic ash clouds and is therefore an important operational consideration. An interesting aspect of this paper is the surprising result that parameterising umbrella spread may actually reduce cloud area in some circumstances (if it reduces the vertical spread of ash and where there is significant vertical wind shear).
I believe this paper can be published after minor revision. My comments are mostly minor points for style and/or clarification.
Line 23. ‘These areas were 2.2, 2.2, and 1.5 times the areas calculated in simulations that ignored umbrella growth.’ It could be made clearer here in the Abstract that simulations which did not include umbrella growth were less accurate (in two out of the three case studies).
Line 43. ‘Umbrella spreading has been incorporated into at least three numerical models used for ash-cloud dispersal (NAME, Fall3d, Ash3d).’ It would be useful here to include some references for the umbrella spreading schemes in each model, and to have a sentence or two detailing previous sensitivity studies on umbrella cloud parameterisation in Ash3d.
Line 53. Again, I am missing some references here (references only given for Calbuco eruption)
Line 55. Minor style point: this is the only section header written in title case.
Line 101. Typo: ‘he’ should be ‘the’
Line 103. Why use equation (3) over equation (4)? Is an increase in u(r) with time more realistic?
Line 113. I don’t think the information in Table 1 needs to be in a table as it contains a single equation and short explanatory note, which would not disrupt the flow of the text.
Line 121. Particle density and shape could also be mentioned here as they are considered later in the paper.
Line 151. As only fine particles (0.01 mm) are considered, and MER is calculated as a function of umbrella cloud growth rate, I assume the MER calculated is just the MER of fine particles (i.e. not total mass eruption rate). It would be useful to explicitly state this.
Line 171. Table 2 shows a single particle size (0.01 mm) was used for Pinatubo. However, the justification for this is only given later in the paper (in the caption to Table 4, which details the inputs to Calbuco simulations, it is explained that the model runs faster with a single grain size and coarser grain sizes are unnecessary). Since the same single grain size is used for all case studies it could be justified earlier in the paper (e.g. around Line 121 where particle size is introduced as an input).
Lines 219-220. It is unclear if wind speed is reduced by 30% to produce the simulation shown in Figure 2c, or whether this was only done for the supplementary figure (which was not provided for review). From Figures 2a and 2c the modelled travel distance seems to match the observed travel distance quite well. If wind speed was reduced in Figure 2c this should be specified when the figure is introduced and preferably another panel included showing the simulation without the wind speed reduction. It is difficult to judge without seeing the supplementary information, but I am not sure what is learned from adjusting the wind field to obtain a better match between model and observations. This seems to suggest that the met data could be inaccurate – there are many other reasons why the particles could have travelled too far, such as uncertainty in eruption source parameters.
Lines 285-286. Typo: ‘…are fit in Figure 4b to two curves to of form…’
Line 357. Figure 5e suggests the non-umbrella simulation produces more accurate cloud area predictions for the Calbuco eruption. The shape of the cloud also looks more accurate for the non-umbrella simulation in this case. These results are in contrast to the other two case studies, where umbrella spreading produces more realistic ash clouds. It would be useful to clarify this in the text and discuss possible reasons for this.
Line 371. The Wilson and Huang shape factor is based on principal axis lengths, and therefore does not take into account surface roughness, which reduces settling velocity. Although a shape factor of 1 is probably technically correct for aggregates, which are usually rounded, you might expect this to give settling velocities which are overestimates given that aggregates generally have very rough surfaces while a shape factor of 1 could also describe a perfectly smooth sphere. In addition, I would use the Wilson and Huang law with caution here given that it is calibrated using a sample of particles which does not include aggregates (and the settling velocity of aggregates is generally poorly constrained given their tendency to break on impact).
Lines 450-451: ‘Addition of smaller or less dense aggregates would produce a distal tail to the deposit that extends farther downwind.’ Following my previous comment above, the drag law used here and the shape factor assigned to the aggregates may be another reason for the mismatch between measured and modelled deposits.
It would be useful to mention the Wilson and Huang drag law and the chosen shape factor of 0.44 earlier if it’s used for all the simulations. It is mentioned in lines 426-427 that the settling velocity of the 10 micron particles is negligible so I would not expect the shape factor to make a significant difference, but it would be good to clarify whether the same drag law and shape factor were used throughout.
Line 392. Some punctuation is missing here.
Line 404. I found Figure 6 took a long time to interpret. In (a) the isopachs are in cm but for the modelled deposits they are in mm. Maybe for the model deposit you could label the 1mm and 0.1mm isopachs so they would be easier to compare with those in (a).
Reviewer 2 Report
Please see the attached comments.

Author Response
Please see the attachment. It includes responses to all three reviewers, including this one.

Reviewer 3 Report
The target of this study is to compare the Ash3d simulation results with the observations from three well-documented large eruptions. Their simulations reveal the consistency and discrepancy in the area covered by the eruption clouds. The logic is clear and the manuscript is well written. The problems picked up in the manuscript is quite important in volcanology and their works have an impact for usage of advection -diffusion models.
I have suggested that this manuscript is acceptable for the publication in Atmosphere.
I would like to give one comment. In lines 336-347, the authors mentioned the mass eruption rate of Calbuco 2015 eruption which was estimated by the previous work. Van Eaton et al. (2016) used equation (1) with lambda=0.2 taken from Suzuki and Koyaguchi (2009). However, Suzuki and Koyaguchi (2009) suggested that lambda is 0.1 in mid-latitude atmosphere, whereas it is 0.2 in tropical atmosphere. The value of lambda for Calbuco eruption could be 0.1 rather than 0.2. I will appreciate that the authors re-estimate the mass eruption rate of Calbuco eruption by using lambda of 0.1.
Author Response

(The authors gave the same response as above.)
